# Efficient Optimization Algorithm-Based Demand-Side Management Program for Smart Grid Residential Load

Ali M. Jasim [1,2,*], Basil H. Jasim [1], Bogdan-Constantin Neagu [3,*] and Bilal Naji Alhasnawi [4]

1 Electrical Engineering Department, University of Basrah, Basrah 61001, Iraq
2 Department of Communications Engineering, Iraq University College, Basrah 61001, Iraq
3 Power Engineering Department, Gheorghe Asachi Technical University of Iasi, 700050 Iasi, Romania
4 Department of Computer Technical Engineering, College of Information Technology, Imam Ja'afar Al-Sadiq University, Baghdad 66002, Iraq
* Correspondence: e.alim.j.92@gmail.com (A.M.J.); bogdan.neagu@tuiasi.ro (B.-C.N.)

**Abstract:** Incorporating demand-side management (DSM) into residential energy guarantees dynamic electricity management in the residential domain by allowing consumers to make early-informed decisions about their energy consumption. As a result, power companies can reduce peak demanded power and adjust load patterns rather than having to build new production and transmission units. Consequently, reliability is enhanced, net operating costs are reduced, and carbon emissions are mitigated. DSM can be enhanced by incorporating a variety of optimization techniques to handle large-scale appliances with a wide range of power ratings. In this study, recent efficient algorithms such as the binary orientation search algorithm (BOSA), cockroach swarm optimization (CSO), and the sparrow search algorithm (SSA) were applied to DSM methodology for a residential community with a primary focus on decreasing peak energy consumption. Algorithm-based optimal DSM will ultimately increase the efficiency of the smart grid while simultaneously lowering the cost of electricity consumption. The proposed DSM methodology makes use of a load-shifting technique in this regard. In the proposed system, on-site renewable energy resources are used to avoid peaking of power plants and reduce electricity costs. The energy Internet-based ThingSpeak platform is adopted for real-time monitoring of overall energy expenditure and peak consumption. Peak demand, electricity cost, computation time, and robustness tests are compared to the genetic algorithm (GA). According to simulation results, the algorithms produce extremely similar results, but BOSA has a lower standard deviation (0.8) compared to the other algorithms (1.7 for SSA and 1.3 for CSOA), making it more robust and superior, in addition to minimizing cost (5438.98 cents of USD (mean value) and 16.3% savings).

**Keywords:** demand-side management; energy management; smart grid; sparrow search algorithm; binary orientation search algorithm; cockroach optimization algorithm; load shifting

**MSC:** 68Wxx

## 1. Introduction

Smart grid (SG) technology is regarded an innovation with the potential to improve the electricity grid in the 21st century. Owing to its distributed generation, universal control, digital two-way communication, and self-monitoring characteristics, the SG has acquired considerable appeal. Using contemporary information and communication technology, the SG can regulate the production of energy, electricity grid distribution, and transportation and develop intelligent monitoring systems. In addition, the SG is capable of managing the power market, controlling decentralized energy resources, and reconstructing infrastructure. Converting the traditional grid to an SG can enable a new era of DSM. DSM can be used to improve grid efficiency, reduce the expense of generation, possibly reduce load pressure, improve system reliability and sustainability, and maximize system capacity

without modifying the power system's physical infrastructure. The concurrently realized objectives of the incorporation of SG and DSM are (i) to minimize carbon emissions in order to combat global warming and (ii) to reduce electricity costs through demand management. By lowering carbon emissions and electricity costs, the combination of DSM and SG can ease the transition of citizens to smart, sustainable, and economic communities [1,2].

Because SGs can be grid-connected or islanded and because the Internet of Things (IoT) is a technique to connect people and things in any place at any time with anyone and anything through any electrical network or service, SGs serve as primary building blocks for an Energy Internet (EI). With the installation of smart meters in residential areas, real-time energy consumption monitoring is possible using EI. EI is hailed as a game-changing network of intelligent grids. It is considered a general IoT application for the energy and power sectors. The EI is made up of a variety of components and techniques, which can be divided into three classes: (i) communication systems, (ii) control algorithms, and (iii) power systems. Electricity generators and users (prosumers) are interconnected with renewable energy resources (RERs), electric loads, and storage systems, opening up infinite opportunities for energy sharing and giving rise to the EI concept. The Energy Internet is a game-changing innovation because it facilitates two-way flows of electricity and data in real time. This change is expected to be caused by the ongoing shift to renewable energy and the improvement of green technologies, such as SGs, storage systems, vehicle-to-grid systems, etc. [3–8].

SG technology facilitates grid connection and RER management and distribution. RERs are intermittent, posing a challenge for the grid. RERs increase the size of abrupt power output deficits due to adverse weather conditions, requiring the grid operator to maintain a higher level of backup power. This may be easily accomplished by reducing energy usage with DSM technology. Thus, load control methodologies must be used. A DSM system ought to be able to communicate with the controllable loads and the main controller [9,10]. The domain of optimal demanded power consumption criteria can be quite broad. Criteria could include increasing distributed production penetration, minimizing peak load demand, and enhancing economic gains by offering customers incentives to reduce demand during peak times [11,12].

The SG is distinguished by its dynamic pricing structures. Dynamic pricing schemes such as time of use (ToU), real-time pricing (RTP), and critical peak pricing (CPP) are frequently used in DSM methodologies, with the main difference being in price levels during operation times. Under RTP, the price changes every hour of the day. Under ToU, prices are fixed in advance (often up to one year in advance), and a variable pricing structure is designed for shoulder, on-peak, off-peak, and low-peak hours. Under CPP, the price of electricity is generally the same throughout the whole year, except during essential peak periods, when it reaches its maximum value. Price adjustments affect only the energy cost outcomes (not energy usage). The utility provides a pricing indication to smart home energy controllers. The energy management controller creates a schedule based on the user's load demand and the price signal. When any dynamic pricing scheme is combined with DSM strategies, the cost of electricity is calculated by the user's energy consumption estimations. Generally, the price can be increased if consumer demand is higher than supply. This growth in electricity pricing impacts all users of the power system. DSM governs the price of electricity in an energy market by lowering the peak demand. To this end, all residential loads are divided into shiftable and non-shiftable categories [13]. During peak hours, DSM techniques modify the demand patterns of customers in order to achieve the desired change in the load shape by shifting shiftable appliances to a more cost-effective time [14]. Therefore, DSM concentrates on energy-saving technology solutions, bill tariffs, and economic incentives instead of improving the grid's transmission and distribution grids or adding more power plants. Moreover, higher consumption demand can cause the load factor deteriorate (average load divided by the peak value), making the system unstable. This can be fixed by rescheduling the distribution system's peak load periods using the proper objective and DSM methodology. The load profile curve can be altered

using six DSM techniques: (1) peak clipping, (2) load shifting, (3) valley filling, (4) strategic load growth, (5) strategic conservation, and (6) load shape flexibility. Figure 1 shows the DSM strategies [15,16].

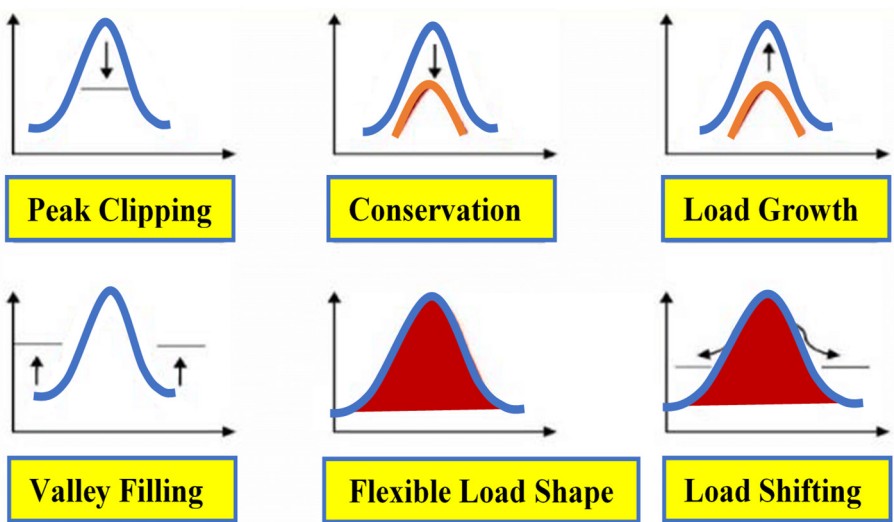

**Figure 1.** Techniques of DSM methodology.

Peak clipping is the practice of removing peaks above a predetermined consumption point. During peak clipping, loads are controlled directly to reduce the pressure of demand during the peak period. This causes a disruption in consumer comfort and compromises the level of consumer satisfaction with the service they receive. To increase electric loads during off-peak hours, valley filling necessitates the use of energy storage units [17]. The primary goal of load shifting is to move on-peak loads to off-peaks periods, thereby lowering the peak demand for energy. By reducing demand directly at the customer location, load profiles can be improved through strategic conservation goals. When there is a high level of demand, strategic load growth enables people to respond more quickly. The load shape greatly affects a smart grid's reliability [18]. In SG management, load control strategies are referred to as flexible loads because they allow for individual participation. Different DSM techniques can be used in a variety of situations, depending on the implementation of the optimizing algorithm.

In this paper, a cost-effective model for residential appliance scheduling is presented. Our appliance-scheduling model seeks to optimize the operational time frame of electrical appliances using the load-shifting technique. The energy generated from SG RERs is considered alongside grid-generated energy in the model. This model simulates ToU pricing and makes use of CSO, SSA, and BSOA to generate optimal schedules. The adopted algorithms are evaluated based on their simple implementation, recentness, and fast convergence. The results show that the proposed model is effective in scheduling the electrical appliances in a residence, which benefits consumers by significantly lowering their electricity bills.

The remainder of the paper is divided into the following sections: Section 2 presents the related work. Section 3 explains the problem statement. The proposed system's overall architecture is discussed in Section 4. Section 5 describes the proposed DSM methodology. Section 6 discusses problem formulation. The adopted optimization algorithms are described in Section 7. The simulation findings are displayed and discussed in Section 8. The paper concludes with Section 9.

## 2. Related Work

In order to reduce energy consumption, peak demand, and carbon emissions, many different methods have been developed to address energy management issues. In this context, the models proposed in [19–22] employ stationary techniques to reduce consumers'

electricity costs and user discomfort. In contrast, [23] presented interactions between interested consumers using a game of repeated energy scheduling, proving that static approaches provide inferior value in terms of both comfort and cost. In comparison, the authors defined non-static DSM techniques in which consumers can choose from a variety of options according to their energy usage and comfort requirements. The DSM architecture is described in detail in [24], in which the proposed design integrates green energy into the power system in order to reduce users' monitoring costs.

The authors of [25] scheduled power and time-shiftable appliances using an integer linear programming (ILP) technique. The authors of [26] proposed a Mixed-Integer Linear Programming (MILP)-based strategy for appliance scheduling to shift loads from peak to non-peak hours in order to reduce peak load and electricity demand costs. In [27], a cost-effective optimization-based model was proposed to control energy use in residences using linear programming (LP) in order to minimize overall cost and the peak-to-average ratio (PAR). An MILP formulation was proposed in [28] to schedule various types of appliances in order to minimize the user's electricity bill. Although dynamic programming (DP) and MILP have been considered to minimize the total cost of running a household and the PAR of electricity use, respectively [29,30], they require a great deal of computing time to implement.

In reference [31], an evolutionary algorithm-based (binary particle swarm optimization (BPSO), cuckoo search (CS), and genetic algorithm (GA)) DSM model was proposed to schedule residential users' appliances, resulting in reduced electricity bills and peaks. In [32], a DSM strategy based on monotonic optimization was presented; the optimal usage of renewable energy was demonstrated by mathematical modeling of a central renewable energy source. Sahar et al. [33] proposed a novel hybrid strategy combining GA, BPSO, and ant colony optimization (ACO) techniques for cost minimization and PAR reduction, taking user comfort into account when pricing ToU services. The authors of [34] proposed a system for managing residential energy demand within the confines of the user's budget. The authors used GA to solve an optimization conundrum with the aim of maximizing user convenience while minimizing energy consumption. The authors of [35] showed how to use RTP to schedule residential loads. To achieve the best electricity use, the authors used fractional programming. Simulated results indicate that the price of electricity was reduced. In [36], the authors came up with a way to shift the electricity load. They used a distributed algorithm to this end. Game theory was used to solve a residential load-scheduling problem. The newton method was also used to speed up the convergence rate of the Nash equilibrium. In [37], a strategy for avoiding distribution system overload was proposed, as well as an algorithm for checking the priority of an appliance and to shut it down to prevent distribution system overload. Overloading of the distribution system is avoided through proper load shedding. The authors of [38] proposed a scheme for scheduling appliances using the optimal stopping rule (OSR), which is a mathematical optimization technique used to indicate the lowest price, allowing the user to schedule their appliance during that time period. This lowers the cost of electricity consumption. In [39], a stochastic cost-minimization problem was proposed, along with renewable energy. The problem of cost minimization was solved using the Lyapunov optimization technique. The authors of [40] proposed a strategy for integrating renewable energy into the power system in order to increase the network's efficiency; users can reduce their monitoring costs by selling/purchasing grid energy. DSM studies were proposed by the authors of [41,42] using GA, PSO, and hybrid particle swarm optimization (HPSO) [43]. A number of engineering applications of artificial intelligence techniques were discussed in [44,45]. In [46], the grey wolf and crow search optimization algorithm was used to create a home appliance scheduling framework. Given the existence of real-time price signals, the proposed method analyzes the cost of electricity savings, user comfort, and PAR reduction for home appliances. To optimize energy usage in homes, in [47], researchers looked into a generic DSM model equipped with a power management controller. Optimization of electricity load scheduling for multiple residents and appliances using a Ladson generalized

bender algorithm was investigated in [48]. In addition, the authors of [49] used a non-dominated sorted GA to schedule home appliances while minimizing the associated energy costs. However, this method is computationally expensive and does not prioritize the convenience of end users.

To assure the lowest energy cost and the highest user comfort, the authors of [50] presented the grey wolf accretive satisfaction algorithm for DSM. In [51], a candidate solution updating algorithm (CSUA) was presented. The goal was to minimize the time a user must wait for PAR and an appliance while still providing that user with the desired level of comfort. By combining the modified and enhanced differential evolution with grey wolf optimization, a model for energy management was proposed and implemented with the goal of reducing peak energy usage and electricity costs [52]. The authors of [53] proposed a strategy for DSM based on load clipping and shifting. This strategy was simulated in MATLAB/Simulink and optimized with an artificial neural network (ANN) algorithm.

In this study, we implemented a smart grid Internet energy-based residential optimal demand management controller using the load-shifting technique. Our implemented model uses BOSA, SSA, and CSO algorithms. Notably, the use of these algorithms for DSM programs has not been mentioned in any previous studies to date, and this is the first study in which these algorithms have been applied in DSM. These optimization algorithms are compared to GA in terms of peak demand, electricity cost, robustness, and computation time tests. The Energy Internet is used to monitor meaningful findings by adopting the ThingSpeak platform. Moreover, adopting on-site RERs decreases peak power plants and reduces electricity costs in the proposed system. Furthermore, a ToU tariff scheme is adopted for electricity bill estimation. The simulation outcomes prove the effectiveness of the energy-optimization controller based on the preceding algorithms. The following are the highlights of the paper contributions:

1. For the first time, an optimal SG residential load-shifting DSM technique based on recent efficient optimization algorithms (BOSA, SSA, and CSO) is been proposed. The proposed DSM model is implemented using ToU dynamic pricing to establish prices in advance, as well as shoulder, on–off-peaks, and low-peak pricing while creating an interactive demand management market in which each consumer plays a role in reducing energy costs.

2. In-home demand consumption can be regulated by integrating applications for embedded systems and the Internet of Things. The model proposed in this study allows for continuous monitoring of the load, as well as scheduling of the load. Adopting EI and the ThingSpeak platform, total energy expenditures and peak energy consumption can be tracked from anywhere in real time.

3. To guarantee the achievement of minimum values of energy consumption, reduced electricity bills, and improved load factor using the load-shifting technique, the adopted algorithms are also compared in term of their robustness (code-tested for 20 times running). Computational speed tests are also performed to determine which algorithm offers the fastest and most effective processing.

4. In order to test the performance and effects of DSM on metrics such as peak consumption and bill electrification with and without DSM, the proposed algorithm-based optimal DSM is compared to the unscheduling load profile and to a DSM program with a commonly used algorithm (GA) for computation and evaluation of the optimal solutions.

5. The proposed optimization algorithm-based DSM program in SG is used to solve the problem of centralized optimization. In particular, each residential load has a local DSM controller and flexible appliances. By optimizing individual scheduling, the energy demand is decreased. The proposed algorithms are simple in construction, require few control parameters, and achieve a high rate of convergence, thereby avoiding getting stuck in local optima.

## 3. Problem Statement

Modern-world concerns include greenhouse gas emissions from fossil fuel electricity generation, requiring electrical researchers, engineers, and policymakers to optimize grid electricity consumption and renewable energy use. Residential loads frequently contribute significantly to both daily and seasonal peak demand, causing the power grid to be over-sized to accommodate peak-period energy usage. DSM allows for more decentralized and efficient operation of appliances through the use of intelligent control strategies, mitigating problems with the current scenario from the perspective of the end user. Numerous other advantages motivate the use DSM; for example, it reduces spending and helps avoid power outages, guarantees a steady and long-lasting flow of power, helps to mitigate environmental concerns by decreasing the need for new conventional power plants, and aids the grid in reducing voltage problems. Without demand management, more power plants will be needed to increase the energy output of the SG and keep up with increasing energy demand. Under optimal DSM, its depth-enhancing benefits are maximized. In our work, an optimal DSM program is proposed to optimally shift the time of shiftable loads and modify the total load of the utility, thereby reducing anticipated peak loads and accomplishing the aforementioned goals. Heuristic algorithms should generally be discovered to find the most appropriate solutions for problems involving global optimization. The chosen algorithms (BSOA, SSA, and CSO) are evaluated based on their simple implementation, recentness (BSOA [54]), and significant advantages, such as their low number of parameters, fast convergence, and immunity to getting "stuck" in a local optimum (SSA [55]). CSO is simple and efficient and has successfully solved global optimization problems [56]. Here, the competency and robustness of the adopted algorithm-based methodology are confirmed. This study paves the way for real-time load monitoring. By using EI and the optimal DSM on the ThingSpeak platform, total energy costs and peak energy use can be tracked in real time from anywhere.

## 4. The Proposed System Structure

Appliances in a residential building should be scheduled in accordance with the ToU pricing model. An automated system must ensure that the workload is properly distributed. Residential energy management (EM) depends heavily on automated appliances, especially in the context of an SG. Below, we present an explanation of the infrastructure and concept of load scheduling in an energy management system.

### 4.1. Model Representation and Concept

Figure 2 shows an illustration of the model's structure, which serves as the basis for the development of optimization algorithms. An integrated power utility is focused on serving a diverse range of loads. To meet peak demand, the optimization program gives preference to residential load appliances that can use power during peak times. This is achieved by shifting schedulable appliances to off-peak hours. As a result, the load-side management system contributes to reducing the energy that is acquired from the utility company.

Smart meters, data centers, a communication network, and data incorporation into application platforms are some of the components of the residential building network. Figure 2 shows a smart meter, which is located between a home or building's local area and utility, and is responsible for transmitting the aggregated demand for electricity to the utility. Smart meters can tell users when and how to use energy, and they can change their habits based on price patterns from the grid. Then, the utility calculates and provides a pricing pattern (e.g., time of use), which is used for load scheduling according to the collected customer data. A distribution board plays a crucial role in any electrical grid. It is used to divide a main power supply into several separate circuits. This board is necessary to separate shiftable appliances from non-shiftable appliances. The smart scheduler (SS) is an EM architecture-integrated device responsible for the scheduling and decision making of smart home appliances. Optimal performance is achieved by combining SS and the appliances. The main power contactor serves as an automatic power switch to transmit elec-

trical power signals from the utility or microgrid (MG) or SG resources to appliances with the help of low-power-relay devices and the distribution board. Lastly, through the smart meter, the adoption of the Energy Internet enables the user to continue real-time monitoring of total energy expenses and peak energy consumption via the ThingSpeak platform.

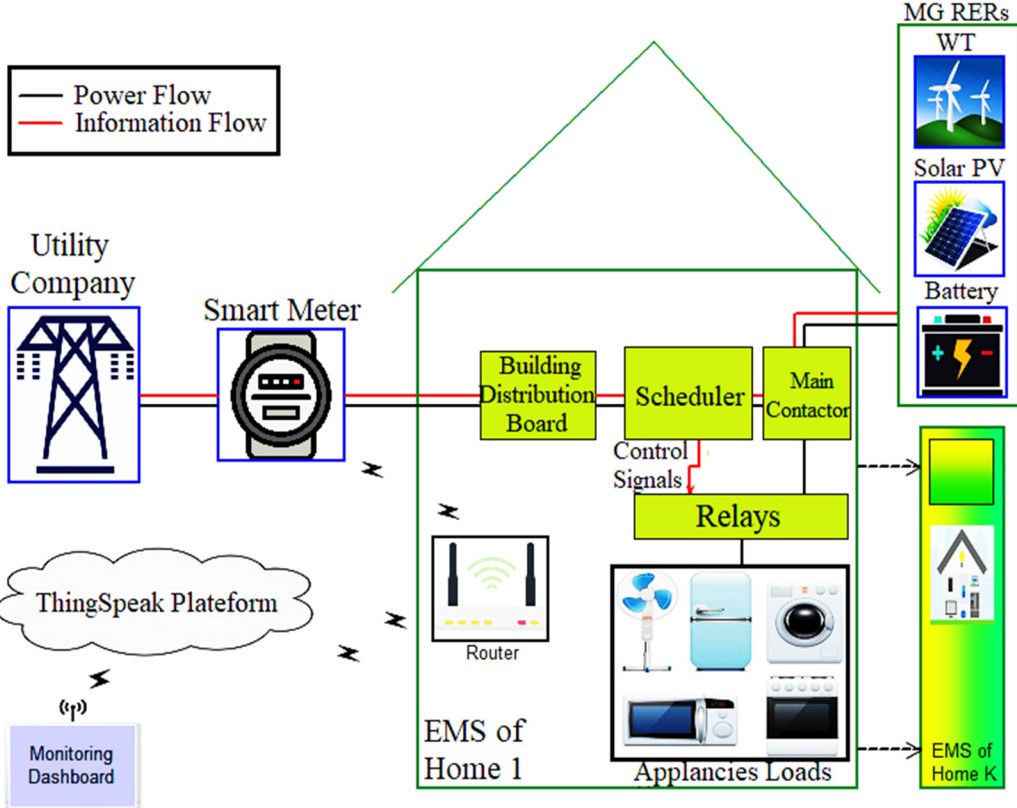

**Figure 2.** Graphical representation of the proposed model.

This work applies an intelligent approach that generates appliance usage patterns based on electricity price tariffs without human intervention. In order to better comprehend energy consumers, we divided them into two main categories: traditional consumers and intelligent consumers. Because traditional homeowners are not concerned with price, they do not include EM architecture in their homes or buildings. EM architecture is not used in traditional homes or buildings, unlike in the homes of smart users, who adopt an EM architecture. The EM system consists of the electrical grid, home appliances, and the display, as shown in Figure 2.

### 4.2. Energy Management System

The home has a smart appliance scheduling and decision-making device, known as a smart scheduler, which is implemented into the EM architecture. SS works in tandem with the appliances. The EMS architecture is depicted in Figure 2. A smart meter sends out energy price signals, as well as a collection of energy-hungry appliances. The SS calculates household appliance ON/OFF schedules in the most efficient way. With a smart meter, the SS receives a signal from the main grid about prices and modifies the user's hourly load demand level in line with the pricing signal. First, the SS moves or shifts the maximum level of electricity usage by each appliance from peak times to off-peak times. The, the SS calculates the cost of electricity for each hour.

### 4.3. Energy Internet

We used a simulation test to monitor the energy demand of the SG according to the Energy Internet approach over the cloud platform to regulate smart home appliances. We

authors created a ThingSpeak platform interface with an effective and simple user interface (UI) that allows homeowners to access and monitor the consumption energy cost and peak energy through cloud-based home energy management. Figure 3 shows an Internet web page that users can access using an Internet browser after entering their username and password as uniform resource locator (URL) login credentials.

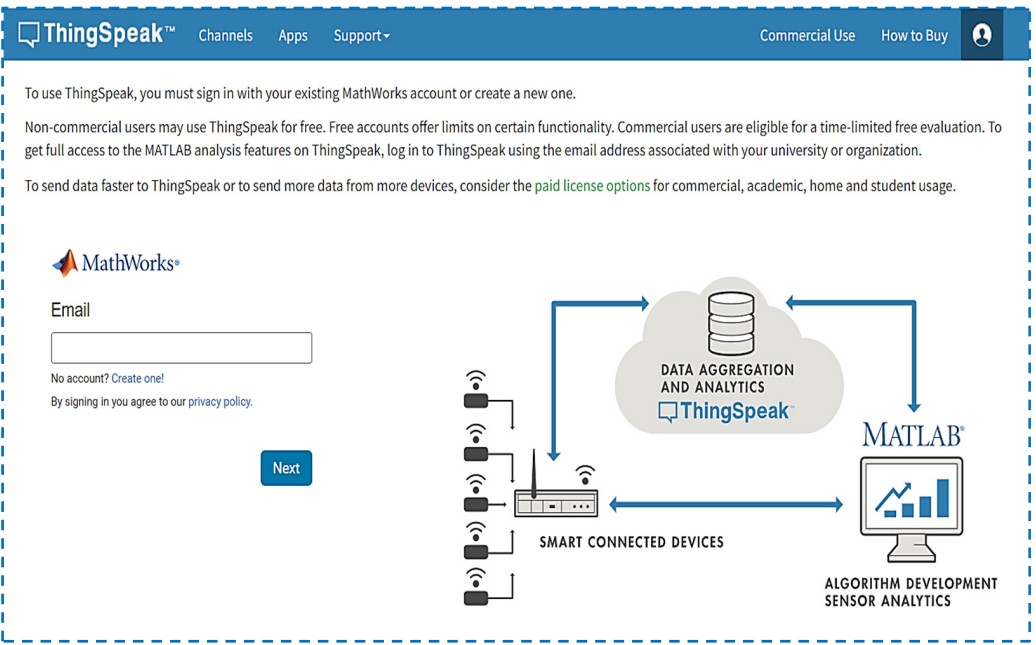

**Figure 3.** Platform for creating user interfaces (ThingSpeak platform).

MATLAB and the open-source IoT framework are used to model proposed communication architectures. For real-time cloud simulation, ThingSpeak was chosen because of the following advantages:

1. Data aggregation, tracking, and analysis on the ThingSpeak Cloud IoT platform. The power profile is graphically depicted and monitored in real time on multiple ThingSpeak channels in the smart grid model.
2. User authentication is enabled by login credentials, and every channel has its own ID. Each channel has two keys for the programming interface. The API's read and write keys are generated at random. These keys enable the storage and retrieval of data from every channel over the Internet or a local area network.
3. A communication network makes it possible for MATLAB and ThingSpeak to send and receive data over the Internet.
4. Data can be imported, exported, analyzed, and viewed on multiple platforms and fields at the same time.

## 5. Proposed DSM Methodology

For purposes of residential load management, we divided home appliances into two categories. The first category consists of shiftable loads, such as vacuum cleaners, washing machines, etc., that can freely be shifted to operate at different times of day without negatively impacting customer convenience. The second type is non-shiftable loads such as electric vehicles, air conditioners, and water pumps, which cannot be operated in different time slots. Table 1 illustrates the rated information of both shiftable and non-shiftable residential loads. There are three water heaters, eight air conditioners, four electric vehicles, and two water pumps. All other appliances have only one unit. Table 2 shows the detailed operation hours and consumption energy of each appliance.

**Table 1.** Shiftable/non-shiftable residential Loads.

| Shiftable/Non-Shiftable Appliances | Appliance Name | Energy Consumption (kWh) |
|---|---|---|
| Shiftable Appliances | Vacuum Cleaner (VC) | 1 |
| | Microwave Oven (MO) | 1.8 |
| | Washing Machine (WM) | 2 |
| | Water Heater (WH) | 3.66 per unit |
| | Dish Washer (DW) | 1.4 |
| | Coffee Maker (CM) | 1.6 |
| Non-shiftable Appliances | Air Condition (AC) | 12 per unit |
| | Electric Vehicle (EV) | 5 per unit |
| | Water Pump (WP) | 4 per unit |

**Table 2.** Detailed operation hours and power consumption of shiftable/non-shiftable loads.

| Operation Hour(s) | VC | MO | WM | WH | DW | CM | Operation Hour(s) | AC (Units) | EV (Units) | WP (Units) |
|---|---|---|---|---|---|---|---|---|---|---|
| 1–2 | ON | ON | ON | OFF | OFF | ON | 1 | 5 | 2 | 2 |
| 2–4 | OFF | ON | ON | OFF | OFF | ON | 2 | 5 | 0 | 4 |
| 5 | OFF | ON | ON | OFF | OFF | OFF | 3–5 | 3 | 0 | 3 |
| 6 | OFF | ON | OFF | OFF | ON | ON | 6 | 2 | 2 | 2 |
| 7 | ON | OFF | OFF | OFF | ON | ON | 7 | 2 | 2 | 1 |
| 8 | ON | ON | OFF | OFF | OFF | ON | 8 | 3 | 4 | 0 |
| 9 | ON | ON | OFF | OFF | OFF | OFF | 9–11 | 8 | 0 | 2 |
| 10 | ON | OFF | OFF | OFF | OFF | ON | 11–13 | 8 | 4 | 0 |
| 11–13 | OFF | OFF | OFF | OFF | OFF | ON | 14 | 8 | 3 | 0 |
| 14 | OFF | ON | OFF | OFF | ON | OFF | 15 | 8 | 2 | 2 |
| 15 | ON | OFF | ON | OFF | OFF | ON | 16–17 | 8 | 0 | 0 |
| 16 | ON | OFF | OFF | ON | OFF | OFF | 18–19 | 2 | 0 | 0 |
| 17 | OFF | ON | OFF | OFF | ON | OFF | 20 | 2 | 0 | 2 |
| 18 | OFF | ON | OFF | OFF | OFF | OFF | 21 | 2 | 0 | 0 |
| 19 | OFF | OFF | OFF | OFF | OFF | OFF | 22–24 | 1 | 0 | 1 |
| 20 | ON | ON | ON | OFF | OFF | ON | | | | |
| 21 | OFF | ON | OFF | ON | OFF | OFF | | | | |
| 22 | OFF | ON | ON | OFF | OFF | ON | | | | |
| 23 | OFF | ON | OFF | OFF | OFF | OFF | | | | |
| 24 | OFF | ON | OFF | OFF | ON | ON | | | | |

A flow chart illustrating the proposed optimal DSM strategy is shown in Figure 4. The first step is to conduct a survey to gather load data. Once the loads have been categorized, a load profile is created, which includes both the shiftable and non-shiftable appliances. Furthermore, the load curve is used to establish the durations of peak and off-peak periods. The amount of energy used in an hour is compared to the maximum allowed for that time period. The load-shifting technique reduces excessive energy consumption by redistributing it among various appliances in use at given time. The technique of load shifting is utilized if and only if the system has shiftable loads. We assume a two-day period to monitor the entire process. Loads can be met by either grid or SG resources once the optimization process is complete. If the energy consumption is less than the total energy of RERs, RERs are used to power the loads. Otherwise, the utility grid compensates for the energy deficit.

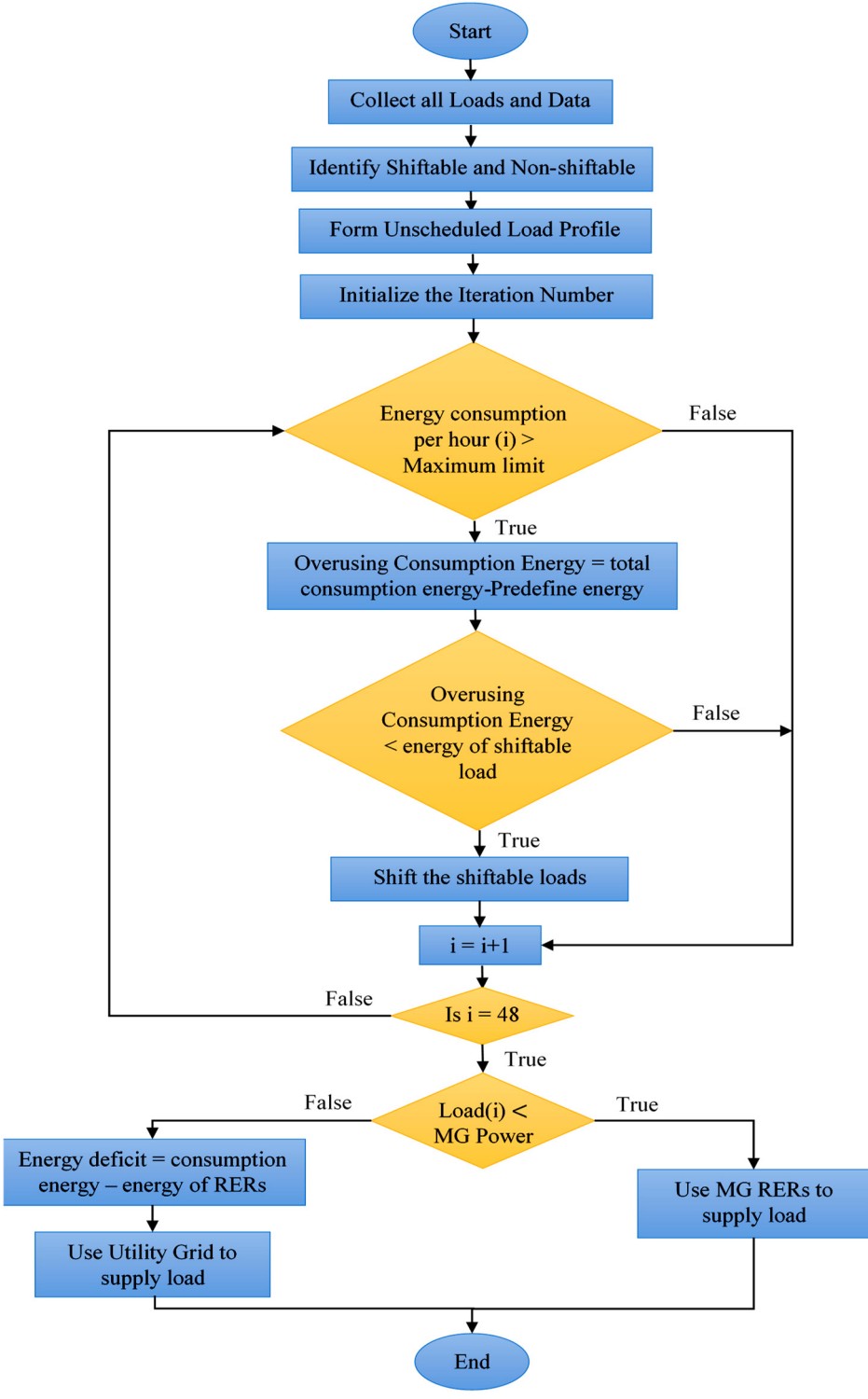

**Figure 4.** Flow chart of the optimal DSM strategy.

## 6. Problem Formulation

### 6.1. Mathematical Framework for Appliance Scheduling

On the basis of energy consumption, end-user preferences, operational hours, appliances can be categorized as either non-shiftable or shiftable. Shiftable appliances can be modified to operate on any time scale without affecting their performance. By shifting their

operations to off-peak hours, energy consumption and costs can be reduced. The daily consumption cost of shiftable appliances ($C_S$) is given by:

$$C_S(t) = \sum_{t=1}^{24} \sum_{m=1}^{N_S} X_S(n,t) \times A_S(n,t) \times PR(t) = \sum_{t=1}^{24} E_S(t) \times PR(t) \tag{1}$$

where $t$ is the time slot, $n$ is the number of appliances, $N_S$ represents the total number of shiftable appliances, $A_S$ is the appliance's power consumption during time $t$, $E_S$ is the total energy consumption of shiftable appliances, and $X_S$ denotes the ON/OFF state of shiftable appliances.

The load profiles of normally operated appliances, which are also referred to as fixed (non-shiftable) appliances and include the AC, WP, and EV, cannot be modified in any way. The daily cost of non-shiftable appliances ($C_{NS}$) can be expressed as:

$$C_{NS}(t) = \sum_{t=1}^{24} \sum_{m=1}^{N_{NS}} X_{NS}(n,t) \times A_{NS}(n,t) \times PR(t) = \sum_{t=1}^{24} E_{NS}(t) \times PR(t) \tag{2}$$

where $N_{NS}$ represents the total number of non-shiftable appliances, $A_{NS}$ is the power consumption of non-shiftable appliances, $E_{NS}$ is the total energy consumption of non-shiftable appliances, and $X_{NS}$ denotes the ON/OFF state of non-shiftable appliances.

The total energy consumption ($E(t)$) and cost ($C(t)$) of all non-shiftable and shiftable appliances are given in Equations (3) and (4), respectively.

$$E(t) = E_{NS}(t) + E_S(t) \tag{3}$$

$$C(t) = C_{NS}(t) + C_S(t) \tag{4}$$

*6.2. Objective Function*

The proposed load-shifting-based DSM schedules shiftable loads so that the energy consumption curve is as close to optimal as possible. Additionally, time slots and movable loads are treated as variable components. Our goal is to minimize the user's electricity bill, in addition to lowering the peak energy consumption to improve the grid's efficiency. The following is the formulation of the minimization problem:

$$Minimize: \sum_{t=1}^{24} E(t) \times PR(t) = \sum_{t=1}^{24} (E_{NS}(t) + E_S(t)) \times PR(t) \tag{5}$$

where $PR$ denotes the electricity price at the specified time ($t$), $X$ denotes the ON/OFF state of appliances, and $E$ is the total energy consumption. The aggregate energy consumption of $N$ appliances during time slot $t$ is equal to or less than the maximum permissible output for energy consumption reduction. An appliance's maximum allowable delay is denoted by $M_n = 24 - l_n$, and the appliance's duration of operation is $l_n$.

*6.3. Constraints*

Constraints should be considered during the process of load scheduling. For example, the total amount of shiftable loads should exceed the total amount of shifted hourly loads. Otherwise, the inflated demand must be reined in. Additionally, shiftable loads have a limit of time shift, which can be advanced or delay within a permissible range. There must be more shifted appliances than there are shiftable appliances at time step $t$, as stipulated in Equation (6).

$$S(n,t) \leq \sum_{t=1}^{24} H(n,t) \,\forall\, -T \leq t \leq T \tag{6}$$

where $S$ and $H$ denote shifted appliances and shiftable appliances, respectively, and $T$ is the limit of time shift.

The load demand for scheduled and shifted loads in the entire day should equal the daily demand usage for loads prior to scheduling.

$$Subject\ to\ \sum_{t=1}^{24}\sum_{m=1}^{M} B(m,t) = \sum_{t=1}^{24}\sum_{m=1}^{M} A(m,t) \tag{7}$$

where $B(m,t)$ is the total daily demand prior to the $t$-th hour of the $m$-th type of load shifting, and $A(m,t)$ represents the overall daily demand after shifting for the $t$-th hour of the $m$-th load type.

## 7. Optimization Algorithms

Three metaheuristic optimization techniques for DSM are covered here. These algorithms are used in typical single building with nine different appliances (six shiftable and three non-shiftable appliances). The energy consumption patterns of various appliances necessitate distinct power ratings. Electricity involves four stages: production, transmission, distribution, and consumption. There are three main types of electricity consumers: households, businesses, and factories. To be clear, our primary objective is to improve the building power scheduling. Many scholars have presented various optimization strategies for DSM in residential areas. As such, we present optimization methods (SSA, BSOA, and CSO) in order to achieve optimal electrical usage. The concept of SSA is inspired by the foraging and predator avoidance behaviors of sparrows. The BSOA is a game-theoretic optimizer based on the principles of the orientation game. Players of BOSA's orientation game, i.e., the searcher agents, move around the playground in response to the direction indicated by the referee. The CSO is an optimization algorithm based on the foraging behaviors of cockroach swarms. Using these algorithms, the shiftable appliances are shifted from peak to off-peak hours by comparing energy consumption with the unscheduled load profile, which helps to bring down the price of electricity because the price goes up gradually as peak use times get closer. The mathematical models and detailed explanations of the adopted algorithms are provided below.

### 7.1. Sparrow Search Algorithm

Xue and Shen presented the sparrow search algorithm in (2020) [57]. The SSA is an algorithm for swarm intelligence optimization. The SSA is based on predator avoidance and feeding behavior of sparrow [57]. It simulates sparrow team foraging; those who seek better food are finders (discoverers), whereas others are followers. Simultaneously, a subset of the population conducts reconnaissance and early warning. If a threat is detected, they forgo food for safety. The matrix below represents the position of individual sparrows [57]:

$$X = \begin{bmatrix} x_{1,1} & x_{1,2} & \cdots & x_{1,d} \\ x_{2,1} & x_{2,2} & \cdots & x_{2,d} \\ \vdots & \vdots & \vdots & \vdots \\ x_{n,1} & x_{n,2} & \cdots & x_{n,d} \end{bmatrix} \tag{8}$$

where $n$ denotes the number of sparrow, and $d$ denotes the dimension of the variable under consideration. Then, the following vector can be used to represent the fitness values of all sparrows:

$$F(X) = \begin{bmatrix} f([x_{1,1} & x_{1,2} & \cdots & x_{1,d}]) \\ f([x_{2,1} & x_{2,2} & \cdots & x_{2,d}]) \\ \vdots & \vdots & \vdots & \vdots \\ f([x_{n,1} & x_{n,2} & \cdots & x_{n,d}]) \end{bmatrix} \tag{9}$$

where $F(X)$ denote the sparrows' fitness, and the value of each row represents a sparrow's fitness. The discoverers are in charge of locating food and aiding the entire population in achieving increased fitness levels while prioritizing food acquisition throughout the search.

Thus, the discoverers can scour a much larger area for food than the participants. When a sparrow spots a predator, it begins singing as a warning signal. This means that if the alarm value exceeds the safety value, the finder directs the group to other secure foraging locations. The updated location of the sparrow finder in each iteration is expressed as follows [57]:

$$X_{i,j}^{t+1} = \begin{cases} X_{i,j}^t \cdot e^{-\left(\frac{i}{\alpha.itermax}\right)} \ for \ R < ST \\ X_{i,j}^t + Q \times L \ for \ R \geq ST \end{cases} \tag{10}$$

where $X_{i,j}^t$ denotes the sparrow finder's location; $t$ denotes the current iteration; $j = 1, 2,..., d$ denotes the dimensions of the i-th sparrow in iteration $t$; $iter_{max}$ denotes the constant with the maximum iterations; $\alpha \in (0,1]$ denotes a random number; $R \in [0,1]$ and $ST \in [0.5,1]$ denote alarm and safety thresholds, respectively; $Q$ is a normally distributed random number; and $L$ is set to 1 if and only if every entry in a dimensioned matrix is a one. $R < ST$ indicates that there are no dangers in the area, so the finder begins a thorough search; $R \geq ST$ indicates that some sparrows have been attacked by predators, and all sparrows must flee as soon as possible for safety.

Individuals with lower energy levels are less likely to forage as part of the group. Some hungry newcomers are more likely to flee in search of additional energy. Entrants can always search for the finder while foraging, as the finder may obtain food or forage in the vicinity. Certain entrants may pay close attention to the finders for increased predation and food competition. Some entrants, on the other hand, pay closer attention to the finders if they notice the finder leaving their current location to compete for food. If they win, they receive the finder's food right away. The following formula is used to update the positions of enrollees [57]:

$$X_{i,j}^{t+1} = \begin{cases} Q \times e^{-\left(\frac{X_{worst}^t - X_{i,j}^t}{i^2}\right)} \ for \ i > \left(\frac{n}{2}\right) \\ X_p^{t+1} + \left| X_{i,j}^t - X_p^{t+1} \right| \times A^+ \times L \ Otherwise \end{cases} \tag{11}$$

where $X_{worst}^t$ is the current worst position in the search space; $A^+$ is a random variable of dimension $d$ with elements randomly distributed between [1,1]; and $A^+ = A^T \left(AA^T\right)^{-1}$. If $i$ is greater than $\frac{n}{2}$, the i-th entrant has a minimal fitness and is most likely to perish. Approximately 10% to 20% of the sparrow population is assumed to be danger-aware, which randomly produces the initial positions of the sparrows. The sparrows on the edge of the group rapidly fly to the secure area to find a better spot. The sparrows in the middle of the group relocate randomly to find other sparrows. The mathematical model of the scout is expressed as follows:

$$X_{i,j}^{t+1} = \begin{cases} X_{best}^t + \beta \times \left| X_{i,j}^t - X_{best}^t \right| \ for \ f_i > f_g \\ X_{i,j}^t + K \times \left(\frac{\left| X_{i,j}^t - X_{worse}^t \right|}{(f_i - f_w) + \varepsilon}\right) \ for \ f_i = f_g \end{cases} \tag{12}$$

where $X_{best}^t$ denotes the current optimal global location; $\beta$ denotes the control parameter for step size in the form of random number normal distribution with a variance of "1" and a mean of "0"; $K$ denotes the direction of sparrow movement in the form of a random number ($\in [-1,1]$); $f$ denotes the fitness function of the optimization problem, where $f_i$, $f_g$, and $f_w$ denote the global current and best and worst sparrow fitness values, respectively; and $\varepsilon$ is the smallest constant required to prevent a zero division error. For simplicity's sake, $f_i > f_g$ indicates that sparrows are at the group's edge, and $X_{best}^t$ indicates that sparrows are around the center of the group; otherwise, $f_i = f_g$ indicates that sparrows in the middle of the population know that there is a threat to their species.

Here, power consumption is managed by introducing the SSA into the DSM control strategy, taking into account the discoverer's position based on the positions of shiftable

loads. The algorithm SSA manipulates the vertical and horizontal axes of the energy consumption pattern. The vertical axis represents the magnitude of energy consumption, whereas the horizontal axis represents the time of energy consumption. This algorithm determines the optimal energy consumption pattern that results in the lowest power cost based on maximum energy consumption and maximum time slot parameters. The chosen objective function to be minimized is shown in Equation (5).

The main steps of SSA are described in the Algorithm 1:

---

**Algorithm 1 SSA Steps**

---

*Step 1:* The utility's ToU price, the daily demand profile, and the unscheduled load timing are all indications of input data that must be defined at the outset of the program.
*Step 2:* Input the control parameters **R, ST, n and** $iter_{max}$.
*Step 3:* Initialize a population with n sparrows using Equation (8).
*Step 4:* Calculate the initial fitness function, and determine the global best sparrow fitness value and global optimal location using Equations (5) and (9).
*Step 5:* t = 1.
*Step 6:* Rate the fitness values and assess the current worst and best evaluation.
*Step 7:* i = 1.
*Step 8:* Update the positions of producers, scroungers, and afraid sparrows using Equations (10)–(12).
*Step 9:* Last individual? yes > return to step 7, else > calculate the updated fitness values.
*Step 10:* If new $x_{i,j}$ less than old $x_{i,j}$ > update the sparrow positions and fitness value, else > return to 7.
*Step 11:* Last iteration?, yes > print the optimal solution, else > return to step 6.

---

### 7.2. Binary Orientation Search Algorithm

BOSA was proposed in (2019) [54] and simulates the rules of an orientation game. In this game, players move around the playground according to the referee's instructions. The starting positions of the players are depicted in Equation (13) [54].

$$X_i = \left( x_i^1, \ldots, x_i^d, \ldots, x_i^n \right) \tag{13}$$

where $x_i^d$ denotes the position of player *i* of dimension *d*, and *n* denotes the number of variables.

In each iteration, the player (P) with the best value of the fitness function is the referee (R), as described in Equation (14):

$$R = \begin{cases} Maximization\ problem : location\ of\ \max(f) \\ Minimization\ problem : location\ of\ \min(f) \end{cases} \tag{14}$$

The value of the fitness function is denoted by (*f*).

A referee's hand may or may not be moving in the same body direction. Players, on the other hand, must only take into consideration the referee's hand. Equations (15) and (16) are used to simulate the direction [54]:

$$P_i = 0.8 + 0.2\frac{t}{T} \tag{15}$$

$$Orientation_i^d = \begin{cases} sign\left(R^d - P_i^d\right)\ for\ rand < P_i \\ -sign\left(R^d - P_i^d\right)\ otherwise \end{cases} \tag{16}$$

At iteration *t* and maximum iteration *T*.

Although each player is required to move in the direction of the referee, a few players may not be able to do so. This problem is simulated in Equations (17) and (18) [54].

$$error = 0.2\left(1 - \frac{t}{T}\right) \tag{17}$$

$$x_i^d = \begin{cases} x_i^d + rand * Orientation_i^d * x_h^d \\ x_l^d + rand * \left(x_h^d - x_l^d\right) \end{cases} \tag{18}$$

where $x_l^d$ and $x_h^d$ are the lower and upper limits, respectively.

In discrete space, the dimensions of the particle position are denoted by the numbers "0" and "1" for each dimension. In any dimension, the movement of an agent corresponds to the change in its value from zero to one or vice versa. Therefore, the displacement in each dimension is determined as a probability function, and the player's position is updated in response to this probability function. In BOSA, the probability function ($dX^{j,d}$) is chosen to be restricted to the interval of [0–1]. The probability function is given in Equation (19) [54].

$$S(dX^{j,d}(t)) = \left|\tanh\left(dX^{j,d}(t)\right)\right| \tag{19}$$

Each player's new position is simulated based on the probability function using Equation (20).

$$X^{j,d}(t+1) = \begin{cases} complement\left(X^{j,d}(t)\right) \; for \; rand < S(dX^{j,d}(t)) \\ X^{j,d}(t) \; Otherwise \end{cases} \tag{20}$$

The following procedures detail how to apply the BSOA-based proposed optimal DSM program to the investigated problem, taking into account the player positions based on the positions of the shiftable loads. This algorithm alters the axes of the energy consumption pattern. The vertical axis shows energy usage, whereas the horizontal axis shows time. Based on the adjusted parameters of maximum energy consumption and maximum time slots, this algorithm calculates the lowest-cost energy consumption pattern. In Algorithm 2, the steps involved in applying BSOA are as follow:

---

**Algorithm 2 BOSA Steps**

---

*Step 1:* The utility's ToU price, the daily demand profile, and the unscheduled load timing are all indications of input data that must be defined at the outset of the program.
*Step 2:* All of the BOSA settings in Table 2 should be set.
*Step 3:* The DSM objective (Equations (5) and (14)) can be minimized by randomly sampling a population.
*Step 4:* The player's position is updated for every population inside the iteration range using Equations (19) and (20).
*Step 5:* Verify each population's constraints.
*Step 6:* Repeat steps 3–5 until the stop condition is met.

---

*7.3. Cockroach Swarm Optimization Algorithm (CSOA)*

CSO is derived from the foraging behavior of cockroaches, which includes swarming, scattering, and light evasion [58–60]. As a result, the CSOA employs a set of rules to mimic the collective behavior of cockroaches. The initial step of the algorithm is to generate a set of potential solutions. Initial solutions are typically generated at random in the search area. Additionally, the CSOA includes three different procedures for the purpose of solving various optimization issues during each iteration, including dispersing, ruthless behavior, and chase swarming. The strongest cockroaches in the chase-swarming process take the best local solutions ($P_i$), create small swarms, and progress toward the global optimum in the new cycle ($P_g$). Each individual ($X_i$) in this procedure reaches its local optimum within its visibility range. Because individuals pursue their local optima in different ways,

it is possible for a cockroach in a small group to be the strongest by finding a better solution. A single cockroach's local optimum exists within its own field of vision, and it seeks the best global solution [61]. Another procedure is for individuals to be dispersed. It is performed on a periodic basis to maintain cockroach diversity. In the search space, each cockroach takes a random step. This process is analogous to the phenomenon of the weakest cockroaches being consumed in the absence of sufficient food [61]. The following is the CSOA model [62,63]:

(1)  *Chase-Swarming Behavior:*

$$X_i = \begin{cases} X_i + step.rand.(P_i - X_i) \ P_i \neq X_i \\ X_i + step.rand.(P_g - X_i) \ P_i = X_i \end{cases} \tag{21}$$

where $X_i$ denotes the cockroach position, $step$ denotes a constant value, $rand$ denotes a random number between 0 and 1, $P_i$ denotes an individual's best position, and $P_g$ denotes the global optimum position. Consider:

$$P_i = Opt_j\{X_j, |X_i - X| \leq v\} \tag{22}$$

where the perception distance, $v$ is constant, $j = 1, 2, \ldots, N$ and $i = 1, 2, \ldots, N$. Consider:

$$P_g = Opt_i\{X_i\} \tag{23}$$

(2)  *Dispersion Behavior:*

$$X_i = X_i + rand(1, D) \tag{24}$$

where the random vector *rand(1, D)* has *D* dimensions.

(3)  *Ruthless Behavior:*

$$X_k = P_g \tag{25}$$

where *k* is a random non-zero integer between [1, *N*].

In this paper, we introduce the CSO into the DSM control strategy to manage power consumption. As a result of this algorithm, the axis along which energy is used is shifted. Electricity consumption is shown vertically, with time displayed horizontally. This method determines the least expensive energy consumption pattern given user-specified maximum energy consumption and maximum time slots. The principal steps for using the proposed optimal DSM program based on CSO are outlined in Algorithm 3 as:

---

**Algorithm 3 CSOA Steps**

---

*Step 1:* Indicators of input data that must be defined at the outset of the program include the utility's ToU price, the daily demand profile, and the unscheduled load timing.
*Step 2:* Set all parameters to their default values and initialize the cockroach swarm using uniformly distributed random numbers.
*Step 3:* Use Equations (22) and (23) to determine Pi and Pg, respectively.
*Step 4:* Use Equations (21), (24), and (25), to carry out chase swarming, dispersion behavior, and ruthless behavior, respectively.
*Step 5:* Loop until a predetermined condition is met.

---

## 8. Performance Results

The adopted algorithm-based system was developed and tested using MATLAB software (R2021b). The simulation program was executed on a laptop computer with an extendable 2.30 GHz processor and 32.00 GB RAM. The program is executed with control parameters of each adopted algorithm, which are illustrated in Table 3. These parameters are set in the following manner: some algorithms (such as SSA and CSO) cannot be stable before 500 iterations (a stable operation means the results are exactly the same at any time of running the code). The maximum intended shift time for every shiftable appliance is 4 h, and this parameter can be adjusted based on the appliance's maximum desired shift time. The maximum energy consumption parameter can be set

to reflect the maximum intended energy consumption of the residential building. The maximum energy consumption for appliances is set at 100 kW. Lastly, the algorithm finds the optimal load pattern by rescheduling shiftable loads within a 4 h window and lowering peak consumption to less than the maximum desired value (100 kW), all based on the user's preferences as set in the algorithm parameters. Figure 5 illustrates the ToU pricing pattern that was been adopted.

**Table 3.** CSO, SSA, and BOSA control parameters.

| Populations Size | Maximum Iterations | Maximum Limit Allow | Max. Shift Time Slot |
|---|---|---|---|
| 30 | 1000 | 100 | 4 |

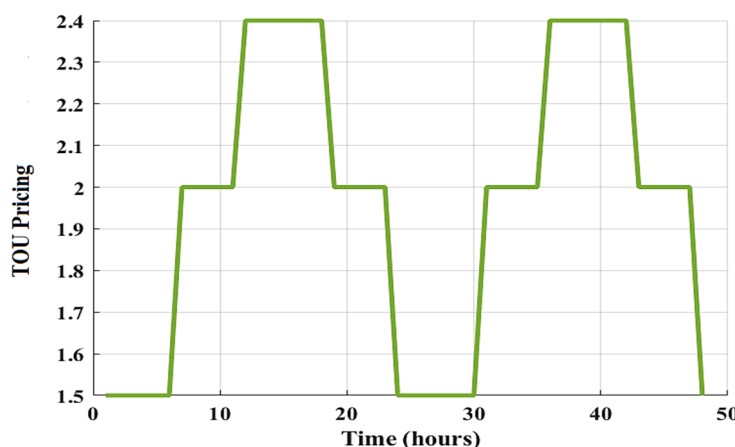

**Figure 5.** The adopted ToU signal.

The non-shiftable load profile on an hourly basis based on the load data (Tables 1 and 2) is shown in Figure 6a. The hourly curve of shiftable loads is presented in Figure 6b. Figure 6c depicts the total load profile, which includes both shiftable and non-shiftable loads. The adopted virtual load data were assumed to be approximately simulated domestic actual load statistics in houses and residential structures. Therefore, they can be installed in homes or residential buildings. When the load data are input into an optimization method, a change is made to the time slot of the unschedulable shiftable load profile. To minimize the energy demand, PAR, and electricity cost, the proposed optimal DSM program only manipulates the non-scheduling shiftable load curve based on the maximum energy consumption and time shift hours according to Table 3. With this change, various appliances may be scheduled to run during off-peak hours rather than during peak hours, resulting in lower energy use. As the time intervals (horizon axis) shift, the magnitude of energy consumption is also altered, as peak consumption is shortened and peak-to-average energy is lowered.

Figure 7 illustrates the simulation results for the adopted residential area energy consumption and for two days (48 h) when the DSM programs based on optimization algorithms are applied and loads are successfully shifted to off-peak hours. It can be seen in the scheduled load curves that on-peak energy consumption (from 9 h to 16 h in the first day) decreases, and off-peak consumption (off-peak hours equal daily hours except on-peak hours) increases. Consequently, the peak-to-average ratio decreases. Therefore, management reduces energy consumption and the electricity cost. The peak demand value of the proposed algorithm-based DSM is up to 87 kW, which is the lower than without the use of the DSM program and optimization algorithms (114.2 kW). Residential customers should attempt to schedule their peak loads for times when electricity prices are relatively low, which results in a lower electric bill. As illustrated in Figure 7, residential users can significantly reduce their daily bill for electricity through proper scheduling of loads. The

adopted algorithms achieve very similar energy consumption and cost-saving results (all up to 16.3% savings). The peak demand is decreased below the maximum predetermined energy consumption limit in the scheduled curve, from 114.2kW to 87kW. In general, the DSM technique performs better as the number of controllable or shiftable appliances grows. The results prove that the proposed optimal DSM program effectively manages a number of residential loads in a residential area by shifting controllable loads in order to reduce peak energy consumption.

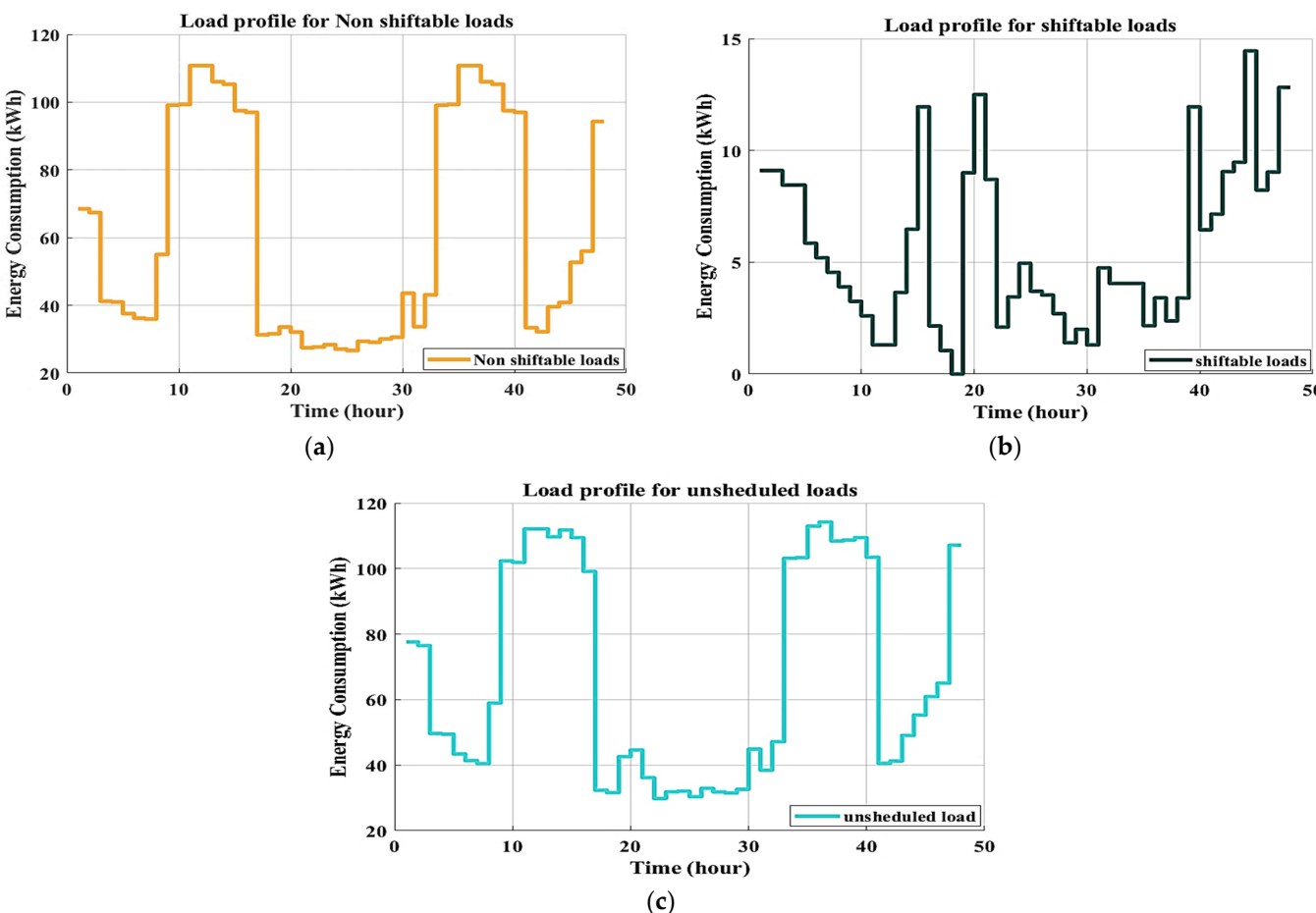

**Figure 6.** Load profile for (**a**) non-shiftable loads, (**b**) shiftable loads, and (**c**) unscheduled loads.

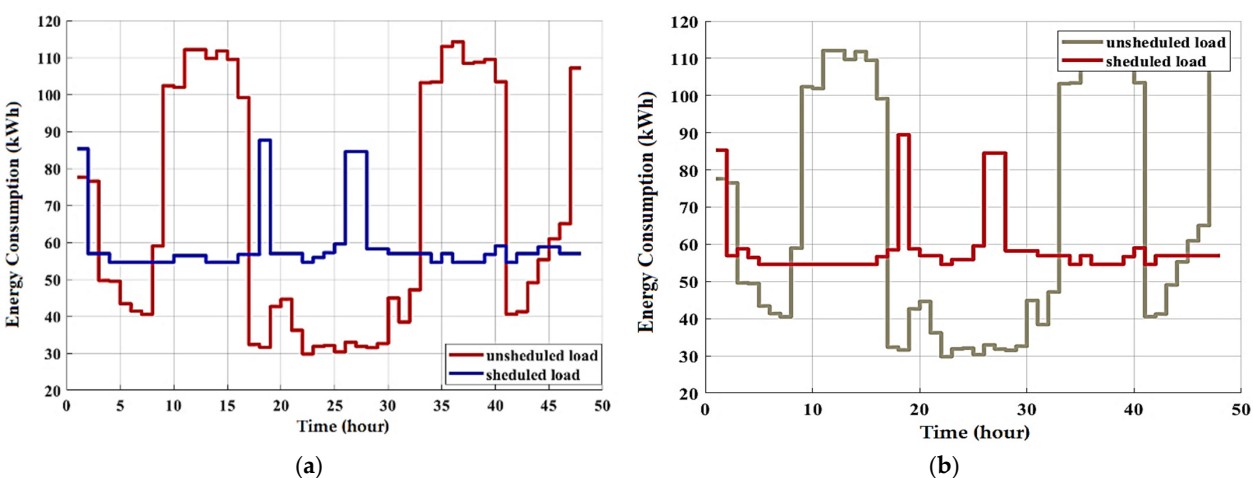

**Figure 7.** *Cont.*

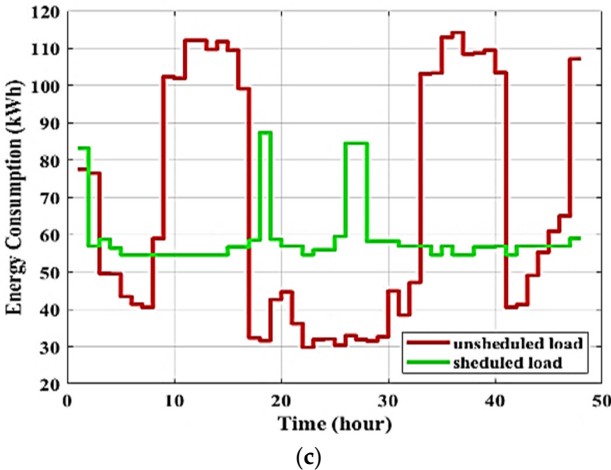

(c)

**Figure 7.** Unscheduled and scheduled loads using (**a**) BOSA (**b**) SSA, and (**c**) CSO.

Figure 8 illustrates the amount of power generated by the MG and the grid, the amount of power consumed by loads, and the costs associated with the grid. Figure 8a,c,e,g assume that the MG supply power is set to 0%, 50%, 100%, and 125% of the total power consumed by the loads, respectively. Their costs are represented graphically in Figure 8b,d,f,h, respectively. As shown in Figure 8a, there is no MG power, and all loads are supplied by the utility grid, implying that the cost will be high, as illustrated in Figure 8b. In Figure 8c, the MG power is half of the consumption power, so there is a power supply deficit. Therefore, the utility grid supplies this deficit, and its cost shown in Figure 8d. There is no power supply shortage in Figure 8e because the MG power is equal to the consumption power. Consequently, the total energy demand is met by MG resources. As depicted in Figure 8f, there are no costs in this case because all loads are supplied by MG resources only. In the final case shown in Figure 8g,h, no power is purchased, but power is sold from the adopted MG to the utility grid, as the MG power is been expanded by 25% of the total consumption power.

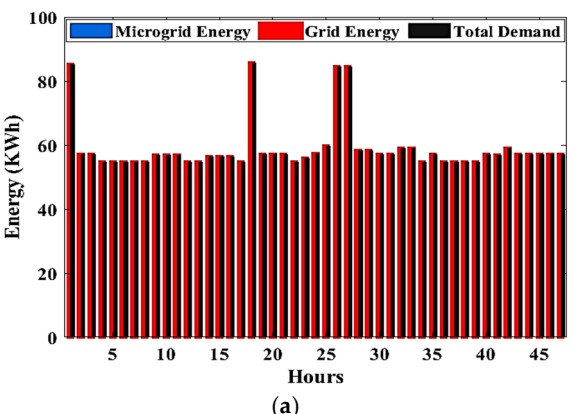

(a)

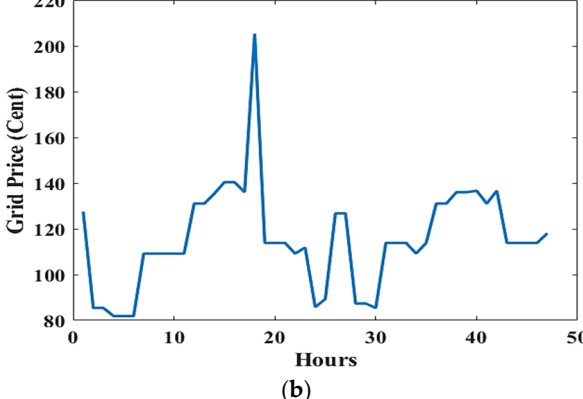

(b)

**Figure 8.** *Cont.*

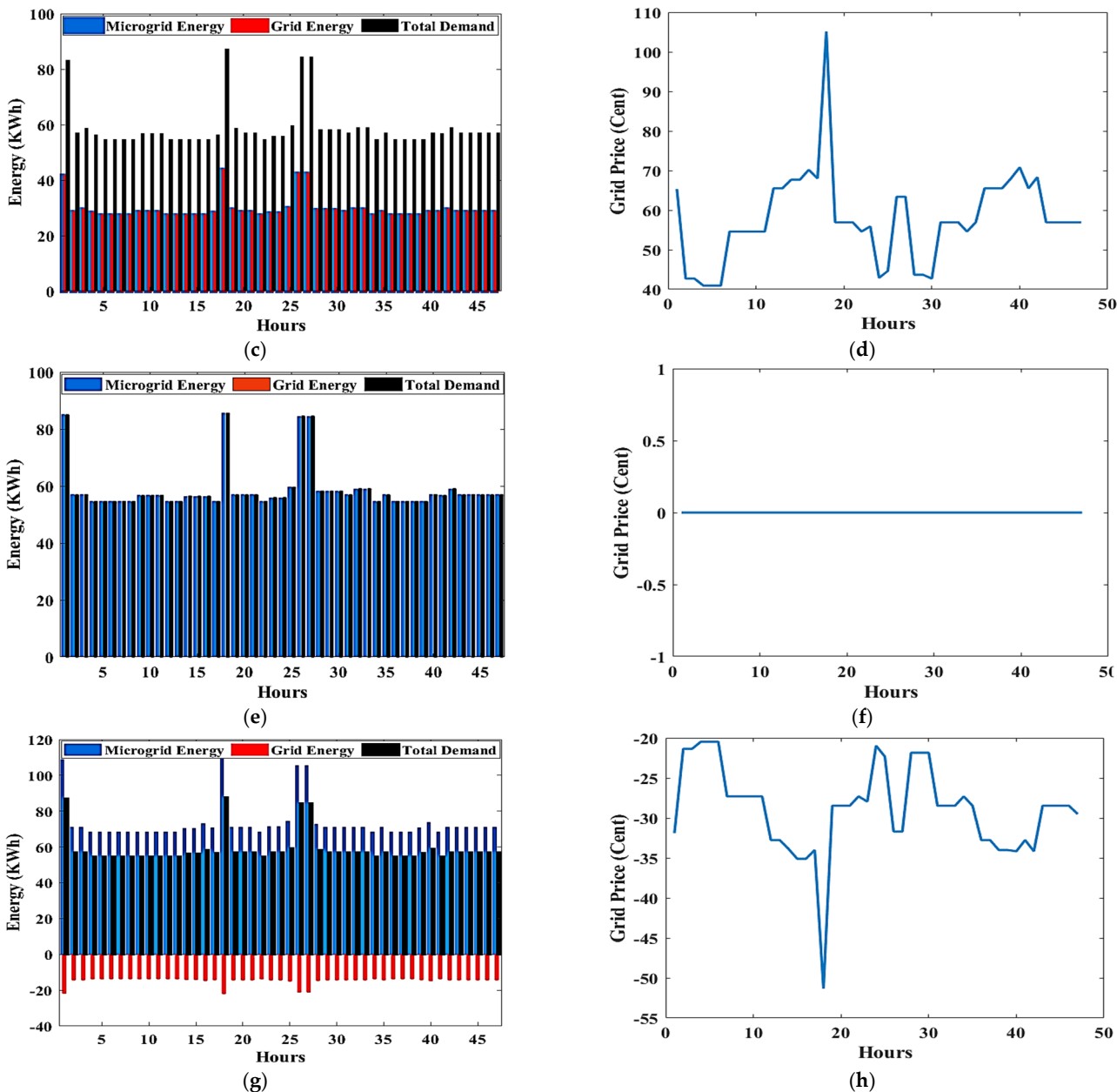

**Figure 8.** The amount of power generated by the MG and the grid, the amount of power consumed by loads, and the grid costs with the MG supply power set to 0 (**a**,**b**), 0.5 (**c**,**d**), 1 (**e**,**f**), and 1.25 (**g**,**h**) from the total power consumed.

Figure 9 demonstrates the peak energy consumption and total electricity cost for both unscheduled and scheduled load profiles, illustrating the outcomes of the adopted algorithms, as well as GA, which is a commonly used algorithm in research studies. It can be observed that all algorithms yield similar results. The unscheduled peak energy consumption and cost are showcased as 114.2 kWh and 650.5 cents of USD, as indicated by black bars. With the proposed DSM, peak consumption energy is reduced to around 87 kWh by all algorithms, except for the BSOA (red bar), which can produce peak consumption of up to 85.8 kWh. The scheduled electricity costs are reduced to 5438 cents of USD. Figure 10 shows the peak amount of energy used and the total cost of electricity for both unscheduled and scheduled load profiles displayed on the ThingSpeak platform using the Energy Internet. As can be seen, the unscheduled peak energy consumption and cost are 114.2 kWh and 6505 cents, respectively. After applying the proposed optimal DSM, the

peak of consumption energy drops to about 87 kWh, whereas the electricity costs drop to 5438 cents of USD. The results indicate that the optimal DSM can properly address the shiftable load in the presence or absence of EI. The convergence rates of the adopted algorithms are shown in Figure 11. The y-axis displays the value of the fitness versus time, whereas the x-axis depicts the iteration number. It is evident that the BSOA converges to the lowest cost compared to the other algorithms (up to 5438 cents of USD). In order to evaluate the robustness of the algorithms, a total of 20 independent runs were performed for each algorithm. Figure 12 demonstrates the mean of peak demand and the standard deviation for a total of 20 runs. As can be seen, the mean value of the CSO is the lowest (86.61 using CSO-based DSM and 114.2 without DSM) in comparison with the other algorithms. Because the BOSA algorithm exhibits the smallest amount of deviation, it is superior to the other algorithms in this regard. It results in peak demand of 85.8 kWh, a cost of 5438.98 cents of USD, and 16.3% savings. Figure 13 illustrates the required computation time for each optimization algorithm. The elapsed time (ET) is computed based on the parameters of each algorithm, as shown in Table 3. It is clear that the BSOA and GA are superior in terms of computation time because they have shorter elapsed times (ET = 27.71 s for the BSOA and ET = 30.22 s for the GA). Finally, it can be stated that the BOSA is superior to the other algorithms in terms of peak energy demand reduction, cost minimization, robustness, and speed of computation.

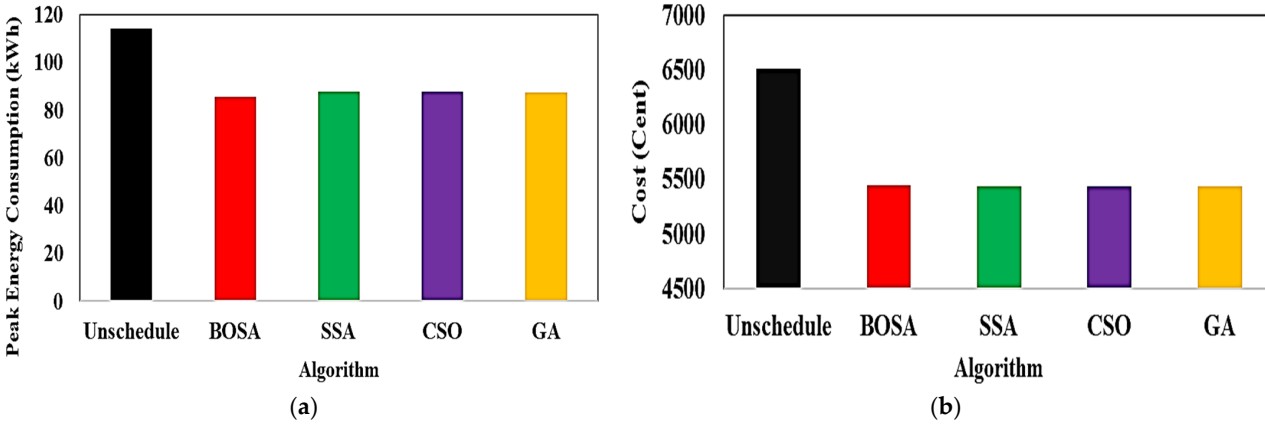

**Figure 9.** (**a**) Peak energy consumption and (**b**) electricity costs for both the unscheduled and scheduled load profiles.

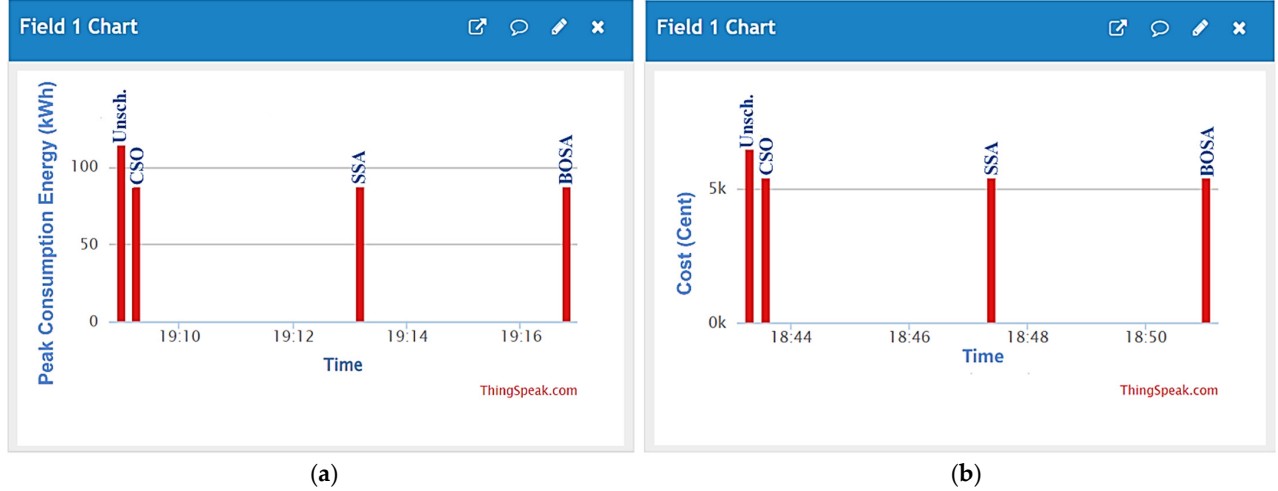

**Figure 10.** (**a**) Peak energy consumption and (**b**) electricity costs for both the unscheduled and scheduled load profiles displayed via ThingSpeak platform.

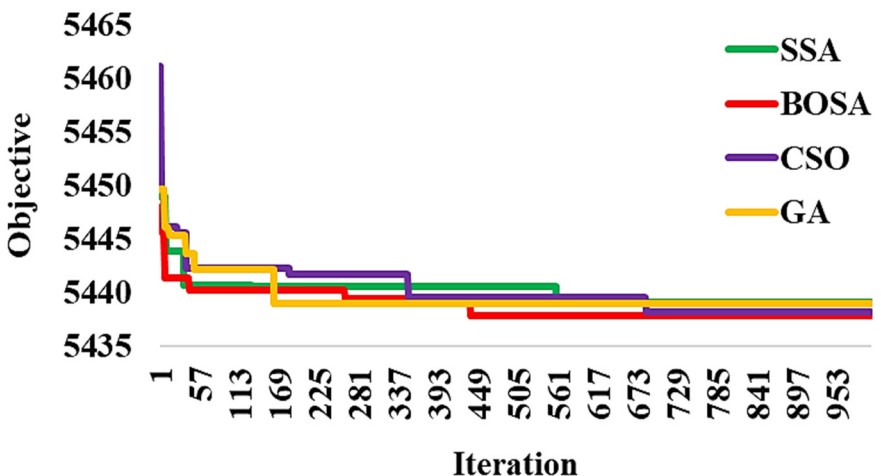

**Figure 11.** Convergence curves for BOSA, SSA, CSOA, and GA.

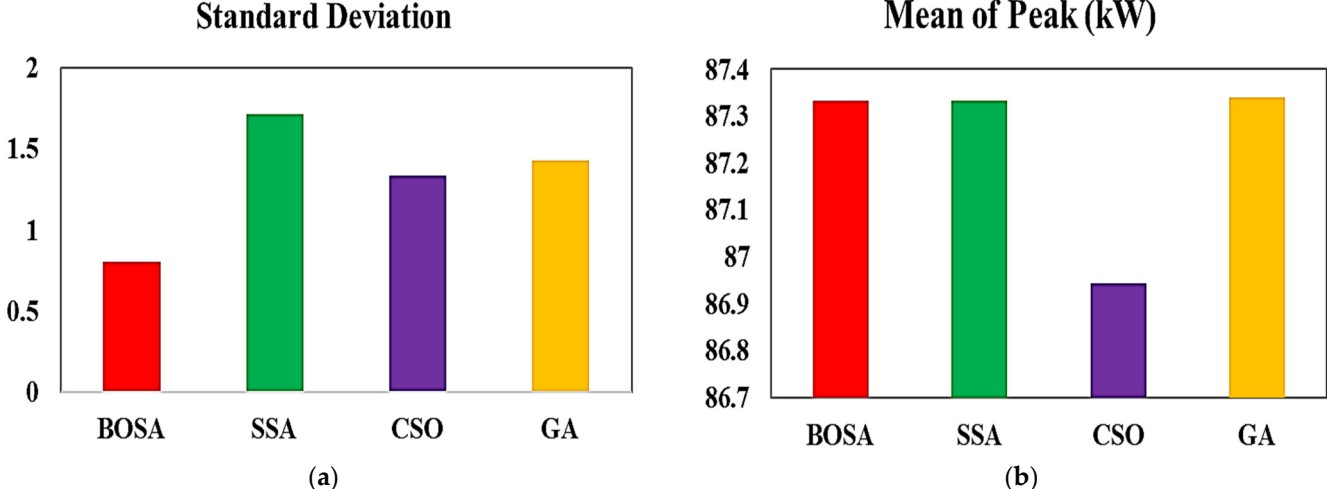

**Figure 12.** Standard deviation (**a**) and mean of peak values (**b**) using BOSA, SSA, CSO, and GA.

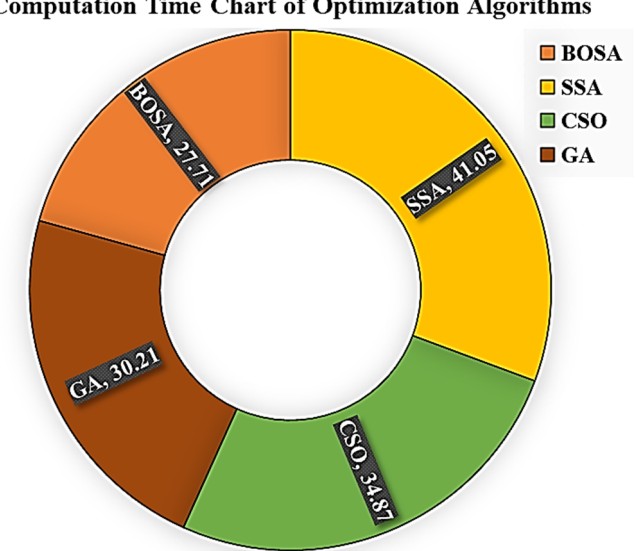

**Figure 13.** Elapsed times of the optimization algorithms in seconds.

## 9. Conclusions

DSM plays a crucial role in ensuring that electricity supply and demand are in balance. DSM aids in the maintenance of a reliable power system and the reduction in both electricity costs and PAR. In this study, we developed MATLAB-based optimization algorithms and an Energy Internet for residential users to reduce peak consumption of the load curve. This work has potential applications in the development of future SGs. Optimal DSM based on metaheuristic optimization algorithms was applied to a variety of residential controllable appliances. The proposed DSM program was optimized by recent optimizers of BOSA, SSA, and CSO using the load-shifting technique. The residential loads are primarily supplied by the SG's RERs, whereas the deficiency is compensated by the utility grid (grid is last priority). In addition, by using secure EI technology, the SG's energies are monitored properly. Total energy expenditures and peak energy consumption can be tracked in real time from anywhere. The proposed model indices, such as peak demand and electricity costs, ensure that the BOSA-based DSM outperforms other algorithms. Whereas the CSO algorithm has the smallest mean value of peak demand (86.61 kWh), the BOSA algorithm has the smallest deviation (i.e., standard deviation for BOSA = 0.8, SSA = 1.7 and CSO = 1.3), making it superior to the other algorithms in terms of electricity costs and savings (BOSA produced 5438.98 cent of USD cost (mean value) and 16.3% savings). Therefore, the BOSA technique is effective in lowering electricity bills and power consumption. Moreover, the results of the proposed approaches were compared to GA results. The GA produces nearly identical results to SSA and CSO in terms of mean peak demand and electricity cost, but the high standard deviation renders the GA inferior. In terms of computation time, the BOSA and GA are superior, owing to their shorter elapsed times (ET = 27.71 and 30.21 s, respectively).

The proposed system has one limitation: it is only applied to controllable appliances in order to minimize energy consumption using an optimal load-shifting DSM technique. It cannot reduce the amount of electricity used by manipulating non-controllable appliances with an optimal peak-clipping technique and operating non-shiftable appliances according to the priority of each appliance. In this paper, we propose recommendations for future research, including, in addition to the load-shift technique, the use of an optimal peak-clipping DSM program and running each appliance according to its priority in order to more efficiently cut power consumption.

**Author Contributions:** A.M.J.: original draft, software, methodology, and validation; B.H.J.: supervision, formal analysis, research resources, investigation, editing, and writing; B.H.J.: validation; B.-C.N.: visualization, project administration, funding acquisition; B.N.A.: editing, validation, and visualization. All authors have read and agreed to the published version of the manuscript.

**Funding:** This research was funded by "Gheorghe Asachi" Technical University of Iasi, Romania.

**Data Availability Statement:** Not applicable.

**Conflicts of Interest:** The authors declare no conflict of interest.

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
