# Peer review of "Efficient Optimization Algorithm-Based Demand-Side Management Program for Smart Grid Residential Load"

_axioms, doi:10.3390/axioms12010033_

Round 1

Reviewer 1 Report

In this document, the authors have applied recent and efficient algorithms such as Binary Orientation Search Algorithm (BOSA), Cockroach Swarm Optimization (CSO), and Sparrow Search Algorithm (SSA) to the DSM program for a residential community, with the main objective of reducing the  the peak energy consumption. Overall, this document addresses an interesting topic and is well organized. But I would like to indicate to the authors some aspects that, in my opinion, they should review and that would improve the document.

(i) Through sections 1, 2 and 3, the authors adequately place the background of the problem they are going to investigate and related articles, relying on abundant references. They also specify the objectives of their research. I think this description greatly helps the reader.

(ii) Section 4 presents in detail the general architecture of the proposed system and res includes two illustrative figures. The only comment is to correct the indentation in subsection 4.1.

(iii) Section 5 adequately describes the proposed DSM program. This presentation is accompanied by appropriate tables and figures.

(iv) The theoretical basis of the investigation and the mathematical formulation of the problem are presented in sections 6 and 7, in detail. In these sections, authors should review in depth two aspects:

- All the elements involved in the equations (functions, parameters,...) must be defined when presenting the equations.

- All the equations must be framed. In many cases it is difficult to read.

(v) Section 7 should be numbered as section 8. In general, simulation results are adequately presented and discussed. But I think that the presentation of the figures in the document should be restructured.

(vi) I think the focus that the authors have given to the conclusions is adequate.

Reviewer 2 Report

The paper is well-written and logically structured. The subject is interesting and the used techniques are also of interest for the scientific community. 

However, the paper needs some polishing before it can be accepted for publication.

1- abstract: The word "program" should be replaced by "methodology"; and the details about the implementation language need to be removed.

2-Related works: The text need to be better organized into some smaller paragraphs, according to the similarity between the reviewed works.

3- Problem statement: I cannot see the utility of this section, which consists of only one paragraph. The justification of the selection of the applied algorithm should be added to the introduction and specifically with the contribution of the work. 

4- Description of the system structure is well structured and easy to follow;

5- Proposed DSM program: do you mean proposed DSM methodology? The usage of the word "program" is not clear;

6- Mathematical formulation is not easy to follow. The explanation of the models defined via the equations needs further justifications;

7- Problem formulation: I suggest to move the definition of the objective function and constraints to section 6. Moreover, due to the ill-formatting of the equations, especially Eqs. 9, 10, 12, 13, 14 and 15, I couldn't grasp well the concepts behind the modeling used therein. Furthermore, Section 7 should be dedicated to explaining the 3 applied algorithms with their instantiation to the problem at hand. Some of the equation in section 7 are also ill-formatted, such as Eqs. 24 and 30. Algorithm description shouldn't be divided between pages.

8- Simulation results, which I think should be termed "Performance results" instead. The text should be re-organized. They are huge paragraph with several aspects. These paragraphs should be structured in smaller ones to improve reading and understanding. The figures do not appear where they should be in the text so they could help following the steps of the evaluation. This is not acceptable. The dashed line around the artworks is ugly and does not help and should be removed. The figures should be better presented: some are with thin bars others with very think bars. The colors used should be associated to the algorithm/metric, etc. They shouldn't be used arbitrarily as they are in the current version. The caption should be kept in the page where the artwork is! I don't understand why the "pizza" diagram is used. A bar diagram is better suited in this case. 

8- The conclusion must also give hints on the limitations of the proposed work.
